# Combined Usage of MDK Inhibitor Augments Interferon-γ Anti-Tumor Activity in the SKOV3 Human Ovarian Cancer Cell Line

**DOI:** 10.3390/biomedicines11010008

**Published:** 2022-12-21

**Authors:** Qun Liu, Jingyu Tan, Zhenguo Zhao, Ruijun Li, Luyu Zheng, Xiangyu Chen, Lina Li, Xichen Dong, Tao Wen, Jian Liu

**Affiliations:** 1Department of Gynaecology and Obstetrics, Beijing Anzhen Hospital, Capital Medical University, Beijing 100029, China; 2Department of Gynecologic Oncology, Beijing Obstetrics and Gynecology Hospital, Capital Medical University, Beijing Maternal and Child Health Care Hospital, Beijing 100006, China; 3Medical Research Center, Beijing Chao-Yang Hospital, Capital Medical University, Beijing 100020, China; 4Department of Orthopaedics, National Cancer Center/National Clinical Research Center for Cancer/Cancer Hospital, Chinese Academy of Medical Sciences and Peking Union Medical College, Beijing 100021, China; 5Department of Oncology, Beijing Chao-Yang Hospital, Capital Medical University, Beijing 100020, China

**Keywords:** ovarian cancer, interferon-γ, therapeutic efficacy, MDK inhibitor, combined utilization

## Abstract

Ovarian cancer (OC) is a particularly lethal disease due to intratumoral heterogeneity, resistance to traditional chemotherapy, and poor response to targeted therapy and immunotherapy. Interferon-γ (IFN-γ) is an attractive therapeutic cytokine, with positive responses achieved in multiple OC clinical trials. However, clinical application of IFN-γ in OC is still hindered, due to the severe toxicity when used at higher levels, as well as the considerable pro-metastatic adverse effect when used at lower levels. Thus, an effective combined intervention is needed to enhance the anti-tumor efficacy of IFN-γ and to suppress the IFN-γ-induced metastasis. Here, we uncovered that OC cells develop an adaptive strategy by upregulating midkine (MDK) to counteract the IFN-γ-induced anti-tumor activity and to fuel IFN-γ-induced metastasis. We showed that MDK is a critical downstream target of IFN-γ in OC, and that this regulation acts in a dose-dependent manner and is mediated by STAT1. Gain-of-function studies showed that MDK overexpression promotes cell proliferation and metastasis in OC, indicating that IFN-γ-activated MDK may antagonize IFN-γ in inhibiting OC proliferation but synergize IFN-γ in promoting OC metastasis. Subsequently, we assessed the influence of MDK inhibition on IFN-γ-induced anti-proliferation and pro-metastasis effects using an MDK inhibitor (iMDK), and we found that MDK inhibition robustly enhanced IFN-γ-induced growth inhibition (all CIs < 0.1) and reversed IFN-γ-driven epithelial-to-mesenchymal transition (EMT) and metastasis in OC in vitro. Collectively, these data identify an IFN-γ responsive protein, MDK, in counteracting anti-proliferation while endowing the pro-metastatic role of IFN-γ in cancer treatment, and we therefore propose the combined utilization of the MDK inhibitor in IFN-γ-based therapies in future OC treatment.

## 1. Introduction

Ovarian cancer (OC) is the most lethal gynecologic malignancy in women and represents the leading cause of gynecologic-related cancer death worldwide, with a five-year overall survival rate of only 29% in patients with advanced disease [1,2,3]. The current standard of care for newly diagnosed OC is cytoreductive surgery and platinum/taxane-based chemotherapy [4,5]. Though about 75% of patients with OC achieve initial responses, the majority of patients experience recurrence and drug resistance within 3 years, and minimal improvement in mortality has been achieved over the past decades [6,7,8]. Therefore, alternative therapeutic modalities complementing surgery and chemotherapy are urgently needed to reduce the recurrence or combat the recurrently refractory disease.

Interferon-γ (IFN-γ) is a promising therapeutic cytokine in the treatment of a variety of cancers, including OC [9]. It is the only member of the type II interferon and functions as a critical orchestrator in the modulation of innate and adaptive immune responses [9,10,11]. IFN-γ exerts antitumor effects by inhibiting cell proliferation, promoting cell apoptosis, and stimulating antitumor immune activity [10,12,13]. In a previous second-line therapy in 1996 [14], intraperitoneal administration of IFN-γ to OC patients twice a week for 3–4 months achieved obvious antitumor responses. In this report, 32% of the 98 assessable OC patients achieved a surgically documented response, of which 23% had a complete response [14]. Moreover, in the first-line chemotherapy of OC in a randomized phase III trial in 2000 [15], the inclusion of IFN-γ led to an improvement of the three-year progression-free survival from 38% to 51% and three-year overall survival from 58% to 74%, with complete clinical responses observed in 68% of patients (versus 56% in controls).

Despite these encouraging clinical benefits, the widespread usage of IFN-γ in OC treatment is still a huge challenge, as major obstacles such as toxic side effects and pro-metastatic adverse effect have also been demonstrated in several reports [10,11,16]. The most common severe side effects are flu-like symptoms, including fever, headache, chills, myalgia, and fatigue, while other side effects include rash, diarrhea, nausea, and leukopenia [9,17]. Besides these side effects, recent advances have also repeatedly indicated that IFN-γ used at lower levels could conversely increase the risk of metastasis in various tumors [18,19,20,21,22]. These adverse effects will doubtlessly reduce the clinical benefits and enhance the risk from IFN-γ-based therapies. Indeed, several clinical cancer trials have reported unfavorable outcomes after IFN-γ treatment. For instance, in a randomized phase 3 clinical trial for advanced OC, patients treated with IFN-γ and carboplatin/paclitaxel showed a significantly shorter survival compared to patients receiving chemotherapy alone [23]. Meanwhile, higher incidences of serious hematological toxicities were more common in the IFN-γ plus chemotherapy group [23]. In addition to OC, there were also studies reporting the non-beneficial outcomes by IFN-γ treatment in melanoma, pancreatic cancer, renal cell carcinomas, breast cancer, leukemia, etc. [11].

Deciphering the molecular basis for cellular response to IFN-γ treatment is thus quite important for improving the tumoricidal function and overcoming the pro-metastatic effect during IFN-γ treatment in order to derive maximal benefits from IFN-γ-based cancer therapies. In the present study, we verified that high-dose IFN-γ exhibited obvious anti-tumor activity by inhibiting cell proliferation, while low-dose IFN-γ exerted pro-tumor activity via activating EMT in OC cells in vitro. We further revealed that the well-known oncogene *MDK* is a crucial responsive gene downstream STAT1 in response to IFN-γ exposure in OC and that MDK is robustly induced by IFN-γ in a dose-dependent manner. Given that MDK significantly promotes cell proliferation and aggressiveness, its stimulation by IFN-γ in OC will inevitably counteract the anti-proliferation but favor the pro-metastasis of IFN-γ, resulting in a reduced therapeutic efficacy. Fortunately, our data indicated that this situation can be reversed by pharmacological inhibition of MDK using a specific inhibitor iMDK.

## 2. Materials and Methods

### 2.1. Cell Lines and Cell Culture

The human ovarian cancer cell line SKOV3 and human embryonic kidney cell line HEK293T were obtained from the National Collection of Authenticated Cell Cultures (Beijing, China). SKOV3 cells were cultured in McCoy’s 5A medium modified (KeyGEN BioTECH, Jiangsu, China). HEK293T cells were maintained in DMEM medium (Gibco, Carlsbad, CA, USA). All the mediums were supplemented with 10% fetal bovine serum (FBS, Ausbian, Australia) and 100 U/mL penicillin–streptomycin mixture (Solarbio, Beijing, China). All cells were incubated at a temperature of 37 °C under 5% CO_2_.

### 2.2. Antibodies and Reagents

The antibody against MDK (1:1000, 11009-1-AP) was purchased from Proteintech (Rosemont, IL, USA). The GAPDH antibody (1:1000, 2118S) was from Cell Signaling Technology (Danvers, MA, USA). The Ki67 antibody (1:500, ZM-0166), anti-rabbit IgG-HRP secondary antibody (1:8000, ZB-2301), and anti-mouse IgG-HRP secondary antibody (1:8000, ZB-2305) were obtained from ZSGB-BIO (Beijing, China). The MDK inhibitor iMDK was bought from Merck Millipore (Darmstadt, Germany). Recombinant human IFN-γ was acquired from PeproTech (Rocky Hill, NJ, USA). The Cell Counting Kit-8 (CCK-8) Kit was purchased from LABLEAD (Beijing, China).

### 2.3. MDK Stable Overexpression

The human MDK lentiviral vector (YOE-LV004-hMDK) and the control vector (YOE-LV004-Ctrl) were purchased form UBIGENE (Guangzhou, China). The MDK and control lentiviral particles were generated in HEK293T cells by co-transfecting the lentiviral vector with the psPAX2 and pMD2G packaging vectors using Lipofectamine 3000 (Invitrogen, Carlsbad, CA, USA). The MDK-overexpressing and control SKOV3 cells were generated by lentiviral infection in the presence of 10 ug/mL polybrene (Beyotime, Shanghai, China).

### 2.4. RNA Extraction and Real-Time Quantitative PCR (RT-qPCR)

The mRNA expression level was detected by qRT-PCR with qPCR SYBR Green Master Mix (Yeasen Biotechnology, Shanghai, China) in a 7500-sequence detection system (Applied Biosystems, Waltham, MA, USA) according to the manufacturer’s instruction. Total RNA was extracted from cells using TRIeasy reagent (Yeasen Biotechnology, Shanghai, China). The mRNA was reversely transcribed into cDNA using the cDNA Synthesis SuperMix (Yeasen Biotechnology, Shanghai, China). Three duplicates were designed for each sample, and the fold change was measured using the 2^−ΔΔCT^ method, with GAPDH as an internal reference. The primer sequences synthesized by Rui Biotech (Beijing, China) were as follows: MDK, 5′-CGCGGTCGCCAAAAAGAAAG-3′ (forward), 5′-TACTTGCAGTCGGCTCCAAAC-3′ (reverse); GAPDH, 5′-AATCCCATCACCATCTTCCA-3′ (forward), 5′-TGGACTCCACGACGTACTCA-3′ (reverse); and Ki67, 5′-AGAAGAAGTGGTGCTTCGGAA-3′ (forward), 5′-AGTTTGCGTGGCCTGTACTAA-3′ (reverse).

### 2.5. Western Blotting Assay

The cells were lysed in RIPA lysis buffer (Beyotime, Shanghai, China) with 1 mM phenylmethylsulfonyl fluoride (PMSF, APPLYGEN, Beijing, China) and 1 mM protease inhibitor cocktail (Beyotime, Shanghai, China). The sample concentration was quantified using the BCA detecting kit (Thermo Fisher Scientific. Inc, Waltham, MA, USA). A total of 40 μg of protein for each sample was boiled at 95°C for 10 min and separated on 10% SDS-PAGE after electrophoresis, and then it was transferred onto a PVDF membrane (Millipore Corporation, Billerica, MA, USA) via a semi-dry transfer unit. After being blocked in 6% skim milk at room temperature for 1 h and washed three times with 1× TBST, the membrane was incubated overnight at 4 °C with primary antibodies and then incubated with anti-rabbit or mouse IgG-HRP secondary antibody for 1h at room temperature. The membrane was visualized using the Baygene Chemilmaging system (Baygene Biotech, Beijing, China) with an ECL kit (Millipore Corporation, Billerica, MA, USA).

### 2.6. Cell Cytotoxicity Assay

The SKOV3 cells were seeded into a 96-well plate at a density of 3000 cells per well and incubated in a water-saturated incubator with 5% CO_2_. The cells were treated with drugs for 48 h before proceeding to CCK8 analysis. A mixture of 10 μL CCK8 reagent, and 90 μL McCoy’s 5A medium was used to treat cells for 1 h at 37 °C in CCK8 analysis. The absorbance at 450 nm was subsequently measured using a microplate reader (Thermo Fisher Scientific. Inc, Waltham, MA, USA). The cell viability (%) = (OD of experimental group − OD of blank group)/(OD of control group − OD of blank group) × 100%. The combination index (CI) for two drugs was calculated using CompuSyn software (version 1.0). A CI value < 1 represents synergism, and a CI value < 0.1 means very strong synergism.

### 2.7. Real Time Cell Analysis (RTCA)

The MDK-overexpressing and control SKOV3 cells were planted in a 16-well E-plate, which was covered with gold microelectrodes in the well, to measure cell proliferation index values indicated by electrical impedance-based detection of cell attachment. The cells were seeded into the E-plate at a density of 2000 cells/well, and they were cultured for about 140 h in an incubator with 5% CO_2_. As the cells grew in the well, the elevated cell index values were plotted as a smooth curve by the xCELLigence System.

### 2.8. Cell Migration and Invasion Assay

The migration and invasion properties of SKOV3 cancer cells were assessed by transwell assays. For transwell migration assay, a total of 5 × 10^4^ cells in serum-free medium were seeded into the upper chamber of a transwell plate (Corning, NY, USA), and the medium containing 10% FBS was added to the lower chamber to serve as a chemoattractant. After 16–24 h, the medium was discarded, and the unmigrated cells remaining on the upper chamber were removed with cotton swabs. The migrated cells were subsequently fixed with 4% paraformaldehyde and then stained with 0.2% crystal violet. Finally, the migrated cells were counted under an inverted bright-field microscope, and representative pictures were taken. Transwell invasion assays were performed following the same protocol but using a transwell plate pre-coated with Matrigel (BD Biosciences, Palo Alto, CA, USA).

### 2.9. Statistical Analysis

The statistical analyses were performed using GraphPad Prism 8.0 (GraphPad Software, La Jolla, CA, USA) by unpaired *t*-test. Data were shown as mean ± standard deviation (SD). A *p* value < 0.05 was considered statistically significant. * *p* < 0.05, ** *p* < 0.01, *** *p* < 0.001, and **** *p* < 0.0001.

## 3. Results

### 3.1. High-Dose IFN-γ Inhibits Cancer Growth while Low-Dose IFN-γ Promotes Aggressiveness in SKOV3 Cells

Despite a promising therapeutic anti-cancer agent, IFN-γ has also been repeatedly reported to have adverse effects in the treatment of various malignances [11,16]. Recent advances reveal that while high-dose IFN-γ leads to tumor regression, low-dose IFN-γ paradoxically increases the metastatic properties of cancer [10,16,24]. To validate the dual roles of IFN-γ in OC, we treated the OC cell line SKOV3 with high-level (1000 ng/mL) and low level (50 ng/mL) of IFN-γ, respectively. Indeed, high-dose IFN-γ caused an obvious inhibition of OC cell proliferation, as indicated by the significantly lower cell density (Figure 1A), as well as a significantly decreased Ki67 level (Figure 1B) in the IFN-γ treated group compared to the control vehicle group. These findings are supported by a previous study showing that continuous exposure of IFN-γ showed direct anti-proliferative activity in five out of six ovarian cancer cell lines in vitro [13] and by another previous report showing that plasmid-mediated IFN-γ stable overexpression significantly decreased the cellular proliferation of SKOV3 cells [25]. However, upon treatment with low-dose IFN-γ for 48 h, the SKOV3 cells, on the contrary, showed an obvious activation of the EMT program by RT-qPCR (Figure 1C) and Western blotting (Figure 1D) examination of molecular markers, as well as the stimulation of cell migration and invasion by transwell assays in vitro (Figure 1E). The similar role of low-dose IFN-γ in triggering EMT and cancer metastasis has also been widely reported in lung cancer [18], melanoma [20], prostate cancer [26], and colon adenocarcinoma [21].

### 3.2. IFN-γ Activates MDK via STAT1 in SKOV3 Cells

With the aim to potentiate the anti-tumor activity and to reduce the pro-metastatic effect of IFN-γ, we first needed to uncover the pivotal vulnerabilities involved in the dual activities of IFN-γ. Here, we found that the well-documented oncogene MDK, which is upregulated in OC (Figure 2A), was robustly induced by IFN-γ in a dose-dependent manner in OC cells at both the mRNA (Figure 2B) and protein (Figure 2C) levels. A significant upregulation of MDK (1.35-fold, *p* = 0.0208) could be detected, even at an IFN-γ dose as low as 1 ng/mL (Figure 2B).

STAT1 (signal transducer and activator of transcription 1) is a major mediator in the IFN-γ signaling that governs gene activation through the binding of the phosphorylated STAT1 to the promoters of downstream genes [27]. To explore whether STAT1 mediates the IFN-γ–MDK axis in OC, we first examined STAT1 responsiveness to IFN-γ treatment and validated that IFN-γ promoted STAT1 expression and phosphorylation in OC cells via Western blotting (Figure 2D) and RT-qPCR (Figure 2E) assays. Subsequently, we blocked IFN-γ-induced STAT1 expression and phosphorylation using a specific STAT1 inhibitor, fludarabine (Figure 2E,F), and confirmed that STAT1 inhibition could remarkably diminished the IFN-γ-induced MDK activation at both the protein and mRNA levels in SKOV3 cells (Figure 2F,G), suggesting that MDK is activated by IFN-γ via STAT1.

### 3.3. MDK Promotes Cell Proliferation, Migration, and Invasion in SKOV3 Cells

To examine the possible role of the activated MDK in IFN-γ-mediated activities, we forced MDK expression in SKOV3 cells, which was confirmed at both the mRNA (Figure 3A) and protein (Figure 3B) levels. We found that forced MDK expression in SKOV3 cells led to a significant increase in the expression of the proliferation marker Ki67 at both the protein (Figure 3B) and mRNA (Figure 3C) levels, suggesting that MDK may promote OC proliferation, which was subsequently validated by real-time cell proliferation analysis (RTCA) (*p* < 0.0001, Figure 3D). These data further indicate that the IFN-γ-induced MDK during OC treatment may counteract the anti-proliferation role of IFN-γ and thus gave rise to a speculation of whether MDK inhibition would help reinforce the anti-tumor effect of IFN-γ by reducing MDK-mediated cell proliferation.

Pro-metastasis is a widely reported adverse effect of IFN-γ treatment [21,26,28]; thus, we further assessed the effect of MDK overexpression on OC metastasis. Transwell assays showed that forced MDK expression significantly drove migration and invasion in OC cells (Figure 3D). EMT activation acts as both a driver and a marker for cancer metastasis [29]. By measurement of the expression of the EMT molecular markers, we found that MDK overexpression resulted in obvious EMT activation, as evidenced by the loss of the epithelial markers ZO1 (Zona Occludens 1) and E-cad (E-cadherin) and the concurrent gain of the mesenchymal markers vimentin and slug, at both the protein (Figure 3E) and mRNA (Figure 3F) levels. The essential role of MDK in mediating OC metastasis was also supported by MDK inhibition analyses using a specific inhibitor, iMDK, which efficiently silenced endogenous MDK expression in SKOV3 cells (Appendix A) and deactivated the EMT program (Appendix A), subsequently suppressing the migration and invasion abilities of OC cells (Appendix A). As there was robust induction of MDK by IFN-γ, even at a very low level, we also speculated that MDK may contribute to the pro-metastasis role of IFN-γ.

### 3.4. MDK Inhibition Potentiates the Tumoricidal Effect of IFN-γ in SKOV3 Cells

To assess whether targeting MDK would augment the anti-tumor activity of IFN-γ in ovarian cancer, we treated the SKOV3 cells with a potent MDK inhibitor, iMDK, and then evaluated the effect of MDK inhibition on the tumoricidal activity of IFN-γ. Though high-dose IFN-γ exerts anti-proliferation function, CCK8 analysis showed that a 48 h treatment with IFN-γ alone caused a only 29.6% inhibition rate at a 5000 ng/mL dose, and only a 34.4% inhibition rate, even at a dose as high as 10,000 ng/mL (Figure 4A); strikingly, combined usage of iMDK at 6.6 nM, a low dose resulting in no obvious cell death when used alone (IC50 = 1238 nm, Figure 4B), led to a robust improvement in IFN-γ anti-tumor activity, causing a 53.9% inhibition rate at only 500 ng/mL IFN-γ and a 58.5% inhibition rate at only 1000 ng/mL IFN-γ (Figure 4C). This indicates that combinatorial MDK inhibition may largely reduce the required therapeutic dose of IFN-γ in OC treatment. The inhibition rate of IFN-γ could be further dramatically enhanced with the increase of the iMDK dose (Figure 4C). Furthermore, CompuSyn analysis showed a very strong synergism between IFN-γ and iMDK (all CI values < 0.1) when combined to treat OC (Table 1).

### 3.5. MDK Inhibition Attenuates the Pro-Metastatic Adverse Effect of IFN-γ

We further assessed the efficacy of MDK inhibition in overcoming low-dose IFN-γ-induced pro-metastatic adverse effect. We treated the SKOV3 cells with 100 nM iMDK, which resulted in an efficient abolishment of IFN-γ-induced MDK activation, as determined by both RT-qPCR (Figure 5A) and Western blotting (Figure 5B) assays. Notably, MDK inhibition significantly abrogated IFN-γ-activated EMT, as indicated by the reversed expression of EMT markers (Figure 5B,C), as well as the attenuated IFN-γ-driven cell migration and invasion in SKOV3 cells in vitro (Figure 5D). Overall, the above data indicate that the combined usage of the MDK inhibitor can not only potentiate the anti-tumor activity but also reduce the pro-metastatic adverse effect of IFN-γ on OC treatment.

## 4. Discussion

Ovarian cancer is the leading cause of death from gynecologic malignancies in women. This disease is usually diagnosed at the late stages with unfavorable outcomes, due to it typically being asymptomatic in the early stage and the lack of effective screening strategies [3,30]. Because of the high post-surgery recurrence in most patients and poor response to targeted therapies and immunotherapies, the survival rate for advanced OC is still poor and has not been substantially changed for over 20 years [31,32].

In the efforts devoted to discover new therapeutic modalities against refractory OC, IFN-γ has been actively pursued for a long history, due to its anti-proliferative, pro-apoptotic, and immune-modulating functions. Accumulating laboratory evidences show that IFN-γ treatment can not only inhibit the in vitro proliferation of OC cell lines but also suppresses the in vivo growth of OC xenografts [12,13,25,33,34,35]. Moreover, previous clinical trials also demonstrated that IFN-γ application led to tumor regression and prolonged the progression-free survival of OC patients [15,36], highlighting the therapeutic potential of IFN-γ in OC.

Despite these early encouraging achievements, the use of IFN-γ in clinical management of malignances is still caught in intense debate because of severe toxicity side effects, pro-metastatic adverse function, and unfavorable outcomes in several reports [10,16,18,24]. These contradictory reports pose a great challenge, and they must be addressed before IFN-γ clinical usage in OC, especially concerning how to improve the anti-cancer activity and suppressing the pro-metastatic effect of IFN-γ. Recent advances indicate that IFN-γ exerts anti-tumor and pro-tumor activities in a dose-dependent manner. At a high therapeutic dose, the high tumoricidal effect may override its pro-tumor effect, leading to tumor inhibition., while at a lower dose, with a relatively weak anti-tumor effect, the pro-metastasis may be a dominating function, leading to tumor progression and metastasis. In support, our in vitro results here also demonstrated that high-dose IFN-γ caused obvious proliferation inhibition, while low-dose IFN-γ led to EMT activation in OC.

Deciphering the molecular mechanism that OC cells exploit to counteract the tumoricidal role and to support the pro-metastatic effect of IFN-γ may help reveal potential vulnerabilities that can be targeted to enhance IFN-γ therapeutic efficacy. The heparin-binding growth factor MDK is a versatile oncoprotein well-reported in conferring cell proliferative and metastatic advantages in various cancers [37]. Recently, we identified, for the first time, that MDK is a critical downstream target of IFN-γ in cancers of various origins, including kidney, lung, cervical, breast, and colon cancers [22]. In this study, we further validated that MDK is also an IFN-γ responsive gene in OC, and this regulation relies on the presence of the IFN-γ response mediator STAT1. Moreover, IFN-γ could significantly upregulate MDK expression at both low and high levels in a close dose-dependent manner. We also confirmed that forced MDK expression in OC could drive cell proliferation and EMT activation. These gave rise to a speculation that OC cells may hijack IFN-γ-induced MDK to counteract IFN-γ-triggered cell proliferation inhibition and to evolve metastatic properties. As expected, the inhibition of MDK by the inhibitor iMDK substantially enhanced the high-dose IFN-γ-induced cell proliferation inhibition and suppressed the low-dose IFN-γ-induced EMT activation and aggressiveness promotion, suggesting that IFN-γ may be used in combination with the MDK inhibitor to improve the therapeutic efficacy and to decrease the pro-tumor adverse effects in OC treatment at the same time.

Albeit these are exciting findings, one of the major limitations of our study is that all experiments were conducted in vitro, which could not mimic the real in vivo tumor context, especially the immune microenvironment. To resolve this, the recent in vitro organ-on-a-chip and advanced microphysiological systems [38,39], which could better capitulate the immune microenvironment, may provide more suitable models. Additionally, more different OC cell lines should be studied, and more in vivo data should be collected before all these observations can reach significance in a clinical setting. Moreover, the IFN-γ-MDK transduction should be further examined in OC patients, and the pharmaceutical combination ratio between IFN-γ and the MDK inhibitor should be further determined in future clinical trials. Though the contribution of MDK inhibition to the immune-modulating role of IFN-γ could not be evaluated under the in vitro background, we still expect a synergistic effect between IFN-γ and MDK inhibition in immune modulation, as MDK has been recently reported to rewire the tumor microenvironment toward a tolerogenic and immune-resistant state, whereas targeting MDK sensitized tumor cells to immune checkpoint blockade treatment [40]. This deserves further investigation in the future.

## 5. Conclusions

Collectively, our findings shed light on an adaptive mechanism exploited by OC cells to counteract the antitumor activity but confer the pro-metastatic effect of IFN-γ by enhancing MDK expression; we further suggest that targeting MDK may be an effective method to augment the anti-proliferation activity of IFN-γ and to attenuate its pro-metastatic adverse effect at the same time, thereby facilitating the usability of IFN-γ in OC treatment.

## Figures and Tables

**Figure 1 biomedicines-11-00008-f001:**
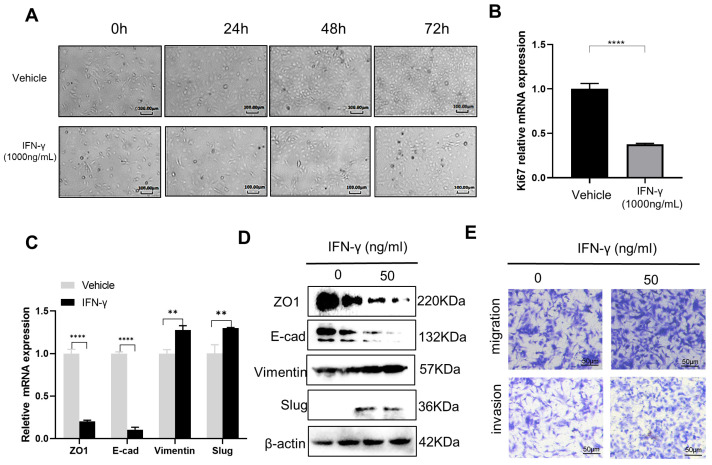
High-dose IFN-γ inhibits cancer growth, while low-dose IFN-γ promotes metastasis in SKOV3 cells in vitro. (**A**) Microscopic observation of the effect of high-level (1000 ng/mL) IFN-γ on cell proliferation of the OC cell line SKOV3. The control cells were treated with vehicle. (**B**) Real-time qPCR to examine the expression of Ki67 mRNA after the SKOV3 cells were treated with vehicle or high-level (1000 ng/mL) IFN-γ for 72 h. (**C**) Real-time qPCR examination of EMT markers after the SKOV3 cells were treated with low-level (50 ng/mL) IFN-γ or vehicle for 48 h. (**D**) Western blotting examination of EMT markers after the SKOV3 cells were treated with low-level (50 ng/mL) IFN-γ or vehicle for 48 h. (**E**) Transwell assays to examine the effect of low-level (50 ng/mL) IFN-γ on the migration and invasion properties of SKOV3 cells. Data are represented as mean ± standard deviation (SD), *n* = 3. ** *p* < 0.01, **** *p* < 0.0001.

**Figure 2 biomedicines-11-00008-f002:**
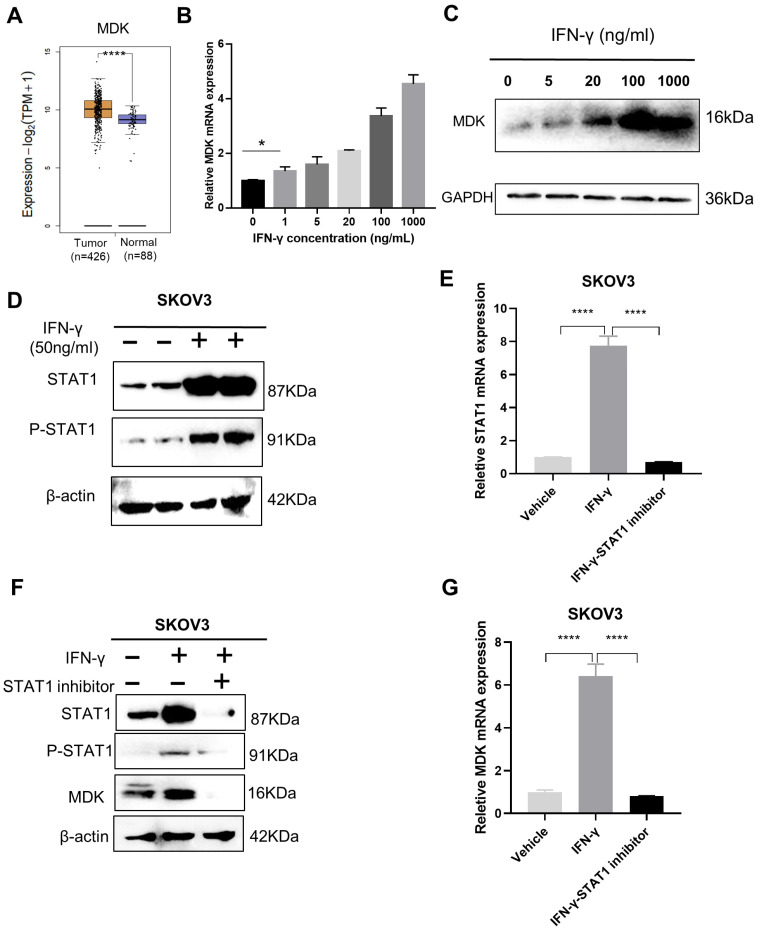
IFN-γ activates MDK via STAT1 in SKOV3 cells. (**A**) GEPIA2 online tool (http://gepia2.cancer-pku.cn/#general, accessed on 6 May 2022) to compare the expression of MDK in the TCGA OV samples and GTEx normal ovary samples. (**B**) Real-time qPCR to examine the expression of MDK mRNA in SKOV3 cells treated with different levels of IFN-γ. (**C**) Western blotting analysis to detect the expression of MDK protein in SKOV3 cells treated with different levels of IFN-γ. (**D**) Western blotting assay to examine the effect of IFN-γ treatment (50 ng/mL, 48 h) on the levels of STAT1 and phosphorylated STAT1 (p-STAT1) in SKOV3 cells. (**E**) Real-time qPCR assays to detect the effect of the STAT1 inhibitor on the mRNA level of STAT1 in IFN-γ-treated SKOV3 cells. (**F**) Western blotting assays to examine the effect of the STAT1 inhibitor on the protein levels of STAT1, p-STAT1, and MDK in IFN-γ-treated SKOV3 cells. (**G**) Real-time qPCR assays to detect the effect of the STAT1 inhibitor on the mRNA level of MDK in IFN-γ-treated SKOV3 cells. Data are represented as mean ± standard deviation (SD), *n* = 3. * *p* < 0.05, **** *p* < 0.0001.

**Figure 3 biomedicines-11-00008-f003:**
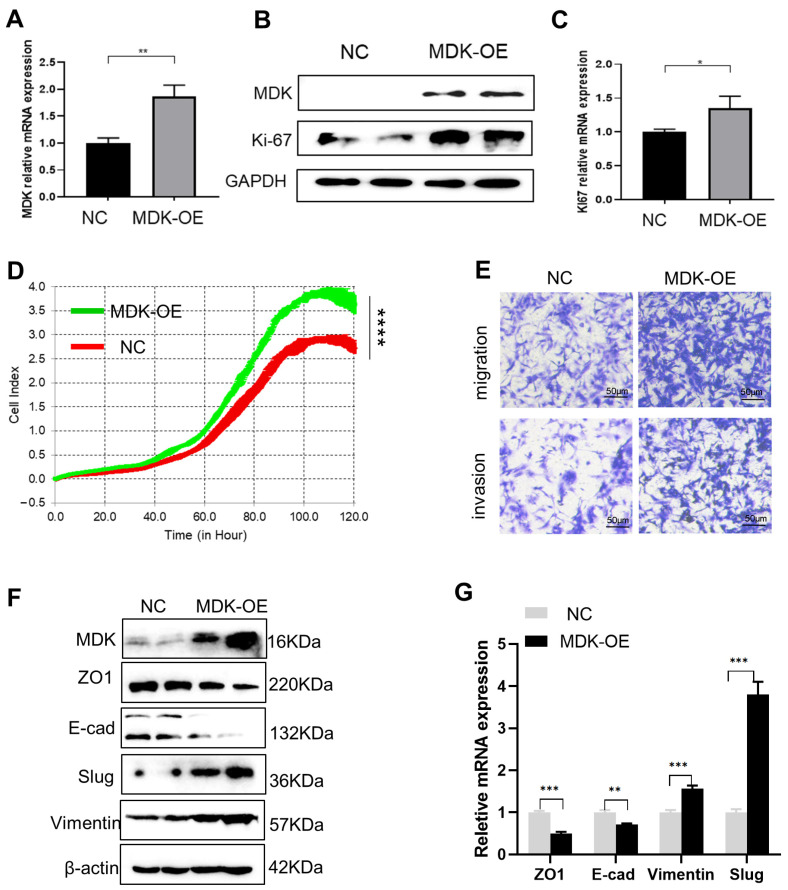
MDK promotes cell proliferation, migration, and invasion in SKOV3 cells. (**A**) Real-time qPCR validation of MDK overexpression in SKOV3 cells. (**B**) Western blotting assays to examine the effect of MDK overexpression on the expression of cell proliferation marker Ki67 in SKOV3 cells. (**C**) Real-time qPCR assays to examine the effect of MDK overexpression on the expression of Ki67 in SKOV3 cells. (**D**) RTCA determination of the effect of MDK overexpression (OE) on the cell proliferation of SKOV3 cells. (**E**) Transwell assays to evaluate the effect of MDK overexpression on the migration and invasion properties of SKOV3 cells. (**F**) Western blotting assays to examine the influence of MDK overexpression on the expression of EMT markers. (**G**) Real-time qPCR validation of the influence of MDK overexpression on the expression of EMT markers. Data are represented as mean ± SD, *n* = 3. * *p* < 0.05, ** *p* < 0.01, *** *p* < 0.001, **** *p* < 0.0001. NC, negative control; OE, overexpression.

**Figure 4 biomedicines-11-00008-f004:**
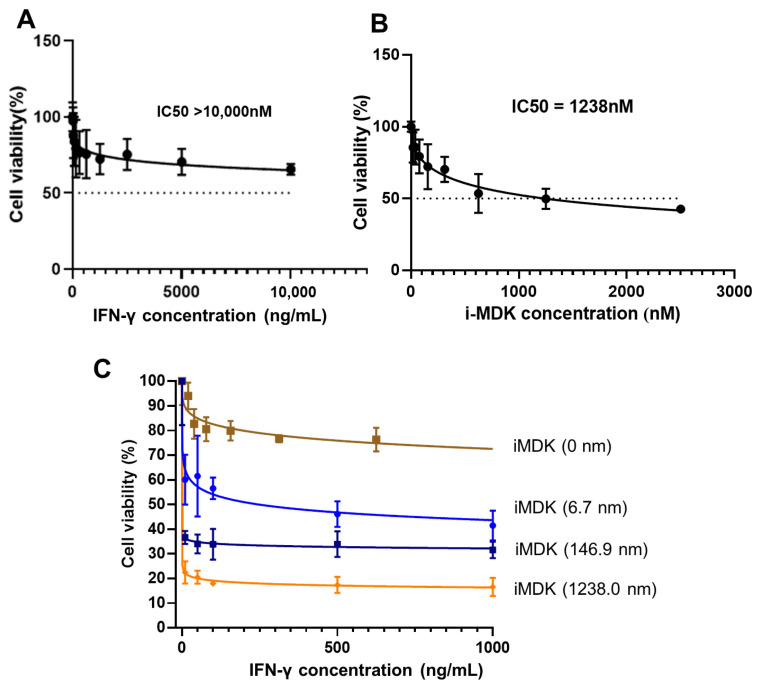
MDK inhibition potentiates the tumoricidal effect of IFN-γ in SKOV3 cells. (**A**) CCK8 analysis of the effect of IFN-γ on the cell viability of SKOV3 cells. (**B**) CCK8 analysis of the effect of iMDK on the cell viability of SKOV3 cells. (**C**) CCK8 assays to determine the effect of the combined usage of different doses of iMDK on the tumoricidal effect of IFN-γ. Data are represented as mean ± SD, *n* = 3.

**Figure 5 biomedicines-11-00008-f005:**
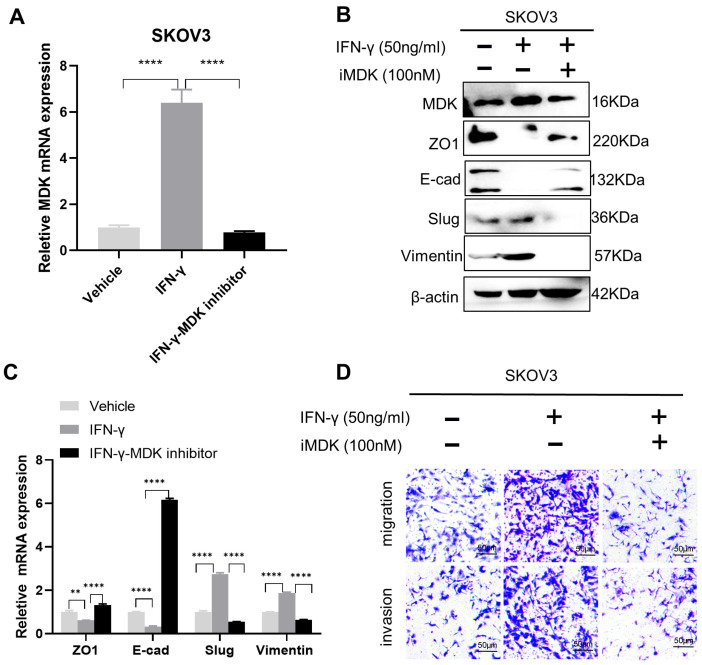
MDK inhibition attenuates the pro-metastatic adverse effect of IFN-γ. (**A**) Real-time qPCR to examine the effect of iMDK on IFN-γ-induced MDK expression at the mRNA level in SKOV3 cells. (**B**) Western blotting assays to detect the effect of iMDK on IFN-γ-induced MDK and EMT markers expression at the protein level in SKOV3 cells. (**C**) Real-time qPCR assays to validate the role of iMDK in suppressing IFN-γ-induced EMT activation in SKOV3 cells. (**D**) Transwell assays to determine the effect of iMDK on IFN-γ-induced migration and invasion in SKOV3 cells. Data are represented as mean ± SD. ** *p* < 0.01, **** *p* < 0.0001.

**Table 1 biomedicines-11-00008-t001:** CompuSyn calculation of the combination indexes between IFN-γ and iMDK.

NO.	IFN-γ (ng/mL)	iMDK (μM)	FA	CI
1	10	6.662	0.3996	0.01426
2	50	6.662	0.3851	0.02502
3	100	6.662	0.4344	0.02188
4	500	6.662	0.5388	0.02493
5	1000	6.662	0.5852	0.02835
1	10	146.896	0.6339	0.03549
2	50	146.896	0.6599	0.02825
3	100	146.896	0.6613	0.02843
4	500	146.896	0.6608	0.03300
5	1000	146.896	0.6845	0.03017
1	10	1238	0.6339	0.06794
2	50	1238	0.6599	0.05325
3	100	1238	0.6613	0.03741
4	500	1238	0.6608	0.03482
5	1000	1238	0.6845	0.03114

## Data Availability

Not applicable.

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
