# Peer review of "Combined Usage of MDK Inhibitor Augments Interferon-γ Anti-Tumor Activity in the SKOV3 Human Ovarian Cancer Cell Line"

_biomedicines, 2022, doi:10.3390/biomedicines11010008_

Round 1
Reviewer 1 Report
General comment:
The authors claim MDK is the IFNg activated pro-metastasis protein, by which mediate the unwanted effect of IFNg for OC therapy. Therefore, suppressing MDK in the IFNg therapy might have therapeutic benefit. This paper is logically reasonable. It’s somewhat pity that this study stop here without conducting more translational approach. Yet, this paper still worth to be documented in the field. Some minor critique raised here.
1. The purpose of combination therapy, in the study, is to block unwanted signaling caused by IFNg. Although author demonstrated the idea indeed work in the cell line system. It’s still no ideally fit to the scenario that combine iMDK did not interfere anti-growth signaling. The data in Fig 4 looks promising that iMDK can even promote IFNg cancer suppressing efficacy. This part is exciting, yet, unexplained.
2. If the iMDK can be a tumor suppressing drug, both suppress growth and migratory activity. Why bother to use IFNg? Why not testing single treatment for OC? There’re several publication using MDK inhibitors for various cancer type, but not yet for OC. Although there is no MDK inhibitor clinical ongoing, it’s still a vivid concept to test that hypothesis.
3. In author’s concept, the iMDK only introduced while IFNg upregulate MDK. However, this left an uncertainty of in what dosage that IFNg induce MDK in patient? What’s the combination ratio pharmaceutically should be determined? This can be taken into consideration in the future clinical trial and put into discussion section.
4. The abbreviation MDK first showed up in the abstract should show full name.
5. The discussion section read very naïve to oncology field. Basically, the author wants to promote the idea of combination therapy, which is fine with the experts in the field. However, please discuss from an angle of more translational state of mind. For example, how the pharmaceutical, toxicological can be conducted, what’s the end points, biomarker suitable for such kind of therapy should be monitored …etc. This information will be useful for the pharmaceutics for further development.
6. However, if you author simple want to claim basic molecular interaction, then please discuss what possibility that iMDK will not interfere (or even boost) anti-growth signaling.
Reviewer 2 Report
This paper is an interesting study of ovarian cancer, and addressing the following comments can improve it for publication
1- The abstract is too long and does not have any quantified results description. It can be improved
2- The scale bars in some of the figures are missing. Figure 1E and others
3- The quality of figures can be improved
4- How about separating the table from the figure 4?
5- It would be great if authors stain cells with live/dead staining to provide an image for cell viability in Figure 4
6- Some of the abbreviations were not defined, for instance, ZO1, … and it would be better to check all
7- Figure S1 should be in the supplementary file
8- MDK should be defined on the first-come
9- The authors should explain how the migration of the cells was tested and provide more details and discussion on that
10- It would be better if the authors can provide more details on experimental methods/assays in the manuscript to be easier for readers to follow the paper
11- In the discussion section, authors can add/mention the potential of organ-on-a-chip and advanced microphysiological systems to provide better in vitro model which can be closer to in vivo models. Here are two relevant references
https://www.mdpi.com/2072-6694/14/3/648
https://doi.org/10.1002/adma.202107876
Reviewer 3 Report
This paper is aimed to shed some light on how the anti-tumor activity of interferon-γ could be increased for the clinical treatment of human ovarian cancer.
This paper describes some in vitro -but not in vivo- obsevations; otherwise, it is technically well designed and performed. It is accompanied by a number of pertinent figures.
The usage of the English language and style is very good.
Specific comments
1. In my view, this paper is essentially an extension of their previous one Zheng L, Liu Q, Li R, et al. Targeting MDK Abrogates IFN-γ-Elicited Metastasis in Cancers of Various Origins. Front 421 Oncol. 2022. 12: 885656 (reference no. 22) but in this case referred to just one ovarian cancer line. And here lies my main concern and reserve as the biology of ovarian cancer is far more complex than examining the in vitro behaviour of just one ovarian cancer cell line. So, I suggest that the title of this paper as “Combined usage of MDK inhibitor augments interferon-γ anti-tumor activity in the SKOV3 human ovarian cancer cell line” would be better reflect its actual content and meaning., as it is the case of Reference no. [25] Lv H, Zhang H, Wu J, Guan Y. Effect of plasmid-mediated stable interferon-γ expression on proliferation and cell death in the SKOV-3 human ovarian cancer cell line. Immunopharmacol Immunotoxicol. 2011. 33(3): 498-503.
2. When describing experiments, the reference to “in the SKOV3 human ovarian cancer cell line” or “in SKOV3 cells” should be used instead of “in ovarian cancer”. Apart from this, the reference to ovarian cancer in a more general context seems to be pertinent.
3. More different cell lines should be studied and more in vivo data should be collected before all these observations could reach significance in a clinical setting. The authors are aware of these limitations as stated in the last paragraph of the Discussion, as well as in Conclusions.
4. In each figure and/or set of data, the n, how many times experiments were done/repeated, should be stated in the text.

Round 2
Reviewer 2 Report
The authors addressed all of my comments